# Effects of Contractual Governance on IT Project Performance under the Mediating Role of Project Management Risk: An Emerging Market Context

**Ayesha Saddiqa** [1,*] **, Muhammad Usman Shehzad** [2] **and Muhammad Mohiuddin** [3,*]

[1] Department of Management Science, Comsats University Islamabad, Attock Campus, Attock 43600, Pakistan
[2] School of Management Engineering, Zhengzhou University, Zhengzhou 450001, China; md38298@gmail.com
[3] Faculty of Business Administration, Laval University, Quebec, QC G1V-0A6, Canada
[*] Correspondence: ayeshasaddiqa2@gmail.com (A.S.); muhammad.mohiuddin@fsa.ulaval.ca (M.M.)

**Abstract:** In this study, we explore the impact of contractual governance (CG) on project performance (PP) under the mediation of project management risk (PMR). Contractual governance influences favorably IT projects performance in an emerging market context where the IT sector is growing. The principal-agent theory is used to build a research model that schedules project governance and IT project risk management. Data were collected from 295 IT professionals and the response rate was 73.75%. Smart PLS was employed to test proposed relationships. The findings postulate a strong causal relationship between the CG, PP and PMR. Fundamental elements (FE), change elements (CE), and governance elements (GE) have a significant positive relationship with project management risk (PMR), and PMR positively affects PP. Additionally, PMR mediates the relationship of FE, CE and GE with PP. Overall, the results of the study provide pragmatic visions for IT industry practitioners and experts, but the unscheduled risk to the IT industry may bring enormous harm. Consequently, effective and well-structured governance in a strategic way tends to improve the project performance by monitoring and managing both project risk and quality. In addition, the study empirically supports the significant impacts of project governance dimensions i.e., fundamental elements, change elements and governance elements on project management risk and project performance. It also guides researchers and adds value to the project performance-related literature by filling the gap.

**Keywords:** contractual governance; fundamental elements; change elements; governance elements; project management risk; project performance

## 1. Introduction

The successful management of IT projects is becoming a major challenge for higher performance [1–3] of software projects. To enhance the performance and efficiency, an effective governance system is needed for project management [1]. Therefore, researchers and practitioners are trying to identify the governance model that improves project management for a better performance. Likewise, project governance is frequently associated with the organizational governance model, which tries to provide comprehensive and continuous methods to control the projects at every phase. Generally, project governance raises" the use of system, hierarchy of procedures and authority to assign sources and organize activities in a project" [4]. In addition, [5] identified that inter-firm IT governance usually consists of contractual and relational governance [6]. This explanation claims that better project governance with its determinants to manage the projects will result in a better project performance.

A lot of work has been done on the relationship between project governance and project performance [7,8]. However, some contradictory prior studies' findings identified inconsistent linkages between project governance and project performance [9,10]. For instance, the results of [11] evaluated that project quality, project management risk and project

governance have extensive influence on project performance, but project governance does not have any significant contribution to enhance project performance. Furthermore, [12] also explored project governance, project risk, and project quality as significant predictors of project performance in the IT sector, and found that the role of project governance is one of the most crucial factors that focus on all aspects of a project containing program management, project sponsors and portfolio management [13]. However, the recent study [1] revealed a significant connection between project governance, project performance and the moderating effects of risk, where a two-dimensional project governance mechanism (contractual governance and relational governance) was used.

Furthermore, previous studies have also proven that project risk and project quality are significant factors that explain a project governance's impact on project performance [11,14,15]. In addition, effective risk management can enhance project performance by focusing the several issues hindering its outcome [11,14]. Effective risk management is a consequence of a sensible project governance system [16]. Additionally, [16,17] stated that project-oriented associations need an exceptional governance system that is different from the customary structures to cope with the risks because the projects are temporary [18,19]. Likewise, [11,20] explored the theory that sensible governance mechanism effectively enhances the quality of a software project, which is a substantial forecaster of project performance. In addition, project quality, including a product's design quality and effective management of project-associated risk, can enhance the performance of software projects [12].

In contrast to this setting, the study in hand explores three-dimensional contractual governance consisting of fundamental, change, and governance elements. Furthermore, [1,11] suggested a focus on contractual governance for the management of project risk and project quality to improve project performance, although the significance of project management risk as a mediator [12,16] in a project governance mechanism has also been tested. However, some of the studies have suggested that the different types of project governance facilitate managing the project risk [12,15] for improving project performance. But the mediating mechanism that illustrates the influence of the contractual governance's elements on project performance has not been investigated. Empirical studies on the mediating role of project management risk in the relationship between contractual governance and project performance are limited. However, most of the methods and models proposed in the literature only discussed the risk management issues of the project itself and ignore the cooperation between the principal agent and the distribution characteristics of the IT projects. Moreover, in recent years, principal-agent theory has been widely employed to solve the problem of risk management of IT projects and good results have been achieved through these studies [21]. To further strengthen our understanding, this study will address the mediating role of project management risk between three-dimensional contractual governance and the project performance relationship using the principal-agent theory.

The main objective of this research is to explore the following research queries; how do contractual governance elements (fundamental elements, change elements, and governance elements) play a vital role in enhancing project performance? Three main purposes have been addressed: (1) to analyze the relationship between FE, CE, and GE of project contractual governance and project performance; (2) to examine the relation between the FE, CE, and GE of project contractual governance and project management risk; (3) to analyze the mechanism of project management risk in project performance and also (4) to evaluate the mediating role of project management risk between the FE, CE, and GE of project contractual governance in project performance.

## 2. Literature Review

The project governance mechanism can be separated into two types: (a) contractual governance which comprises fundamental elements, change elements, and governance elements and is mainly focused on conforming to official rules and highlighting the value of contracts and written agreements related to the deals between the two groups [22–25]; and (b) relational governance which comprises trust, information exchange, solidarity,

and flexibility and is an unceremonious type of governance with emphasis on making friendly contacts between the parties involved in a corresponding transaction [24,26]. Governance elements are used to claim and then conclude an agreement, and settle disputes between parties [27]. In addition, to perceive the elements of contractual governance, [28] highlighted that the logic of the fundamental elements of contractual governance is to circulate the shared views of both parties involved in a contract so that their IT outsourcing deals could be established due to their common objectives and a general commitment. Moreover, effective management of risk is a consequence of a sensible project governance system [16]; therefore, change elements of contractual governance deal with provisions specifying codes, measures, tactics, structures of organization and processes of resolving unexpected happenings or risks [27].

### 2.1. Principal–Agent Theory

The relationship between the principal and the agent is a procedure in which the agent performs his/her duty of managing the firm on behalf of the principal. In this role, the agent acts on behalf of a specific principal and should not have a conflict of interest in carrying out the act [29]. This study examines the project governance through Michael and William's theory of principal–agent. The theory was established in the 1970s and is a widely renowned approach in the literature of project governance. It emphasized the need for a categorical contract between the principal and the agent. Additionally, this theory claims that it is ineffective that directors take responsibility directly for the management of any organization [30]. The theory also highlights the subsequent goal clash that arises when people with different preferences participate in a cooperative effort. Hence, the use of the principal–agent theory results in the partition of ownership from management control [31]. Moreover, on the basis of this theory the principal hires an agent to manage the firm on his/her behalf [32]. Likewise, the concept of accountability is the fundamental component of the principal–agent theory. Furthermore, due to the generalized context of principal–agent theory as it is applicable in project management, it has also been usefully comprehensive to the project environment.

### 2.2. Contractual Governance (CG)

A structure of governance as expressed by the OECD (2004) seeks to decrease clashes between different stakeholder groups which can negatively affect project performance. Contractual governance is all about delivering business benefits, well-managed services, and a strong customer–supplier relationship. It also offers a framework by which the organization's objectives are fixed. Furthermore, governance has two types according to prior literature [27], i.e., "change governance" and "contractual governance", which highlight the significance of compliance's formal rules and contracts among the business partners [23]. Additionally, contractual governance emphasized the use of a formal and legally binding agreement to specify the partnership of the inter-organization transaction [25]. Moreover, [22] the contractual governance is a deal as the principal source for safeguarding dealings.

The fundamental elements (FE) of contractual governance in IT projects explain the key rules and contracts between the groups and parties, including explaining their duties and responsibilities. Moreover, fundamental elements generally describe the contracts in terms of supply deadline, quality standards and project budget [27,33]. In addition, [28] the logic of the fundamental elements of contractual governance is to circulate the shared views of both the parties that are involved in a contract, so that the IT outsourcing deals by which they carry out their basic needs to run their projects could be established through their common objectives and a general commitment.

The change elements (CE) of contractual governance include the processes to resolve unforeseen consequences for future claims, the processes for the execution of foreseeable probabilities and variations, the processes of innovations matched to inducement plans as well as the processes to boost the response and efficiency modifications [33]. Moreover,

they deal with provisions specifying codes, measures, tactics, organization structures and processes of resolving unexpected happenings [27]. Earlier studies on external IT projects further emphasized the feasibility and impact of uncertainties or evolutionary norms of investigating unstructured works [28].

The governance elements (GE) of contractual governance focuses on how relationships are defined by the metrics, forfeits and incentives, options and responsibilities for retirement, the process of documenting communication and the identification of potential disputes as well as how to resolve them [33,34]. Moreover, governance elements identify methods to preserve the relationships through a clear statement of the measurements, punishments and incentives, to claim and to conclude an agreement and to settle disputes between parties [27]. Consequently, the contractual features of governance laid down the administrative processes [35].

### 2.3. Project Management Risk (PMR)

Risk is known as the unseen or uncertain situation. A project is unique by definition and hence during the execution it deals with anonymous issues that are defined as the risks of the project [36]. A general risk management process has the following steps: 1—context establishment; 2—risk identification; 3—risk assessment, measuring the level of risk; 4—risk calculation; 5—risk control/treatment [29]. Moreover, risks in the project can negatively affect and create hurdles to achieving the objectives of that project [17]. Project risk discusses that condition, which poses a severe threat and makes the barrier to completing projects relevant to the IT field [15]. According to [37], risks and uncertainties are integral to the projects. Though it is impossible to fully remove risk from any business transaction, it is, however, possible to manage by investing in managerial expertise and resources in an organizations. Furthermore, several approaches that are used to manage the project risk follow the logic of project process groups all through the life cycle of a project, which needs to use numerous techniques and tools to manage it [38,39]. However, several authors [40], claimed that many studies, risk management strategies and different approaches to risk management are required.

### 2.4. Project Performance (PP)

Project performance (PP) terminology is used to define the presentation of the project, which is measured by different parameters such as time, cost and quality [41]. Likewise, [8] reported that interest in project performance was developed in 1996, for the first time. The concept of an iron triangle, introduced by Barnes in 1969, which includes time, cost and quality, used to managing the "quality" besides time and cost [42] to measure the performance of the projects that can be considered during the execution of the project. Hence, the level of performance of the project is measured in terms of compliance with the cost of the project, the timetable, the objectives, technicalities, quality and profit [43]. It is established [44], a model in the field of project management practice which developed through a ratings system that has eight classifications: four of them are related to measuring the project performance through cost, quality, schedule, and process efficiency. Moreover, [45] introduced the conceptualization of project performance, which contains the product's performance and the process. Furthermore, by explaining the product performance, it discusses the effectiveness of the already developed system, while process performance itself deals with the effectiveness of the development process, which means to what extent the project was carried out according to schedule and budget [14].

### 3. Hypotheses Development

#### 3.1. Contractual Governance, Project Management Risk and Project Performance

The planning of the several project practices (that are applicable to different organizations through communication to drive a project) are generally administered by a contractual relationship, and the contracts are eventually selected by a governance mechanism [27]. According to [1], contractual governance and its foremost elements performed as a significant

interpreter of an effective project performance. As per the definition of the Project Management Research Committee of China in 2009, a contract in a project is an agreement reached by the participants, containing natural people, authorized entities and further organizations as well. By acknowledging [46] the worth of the contractual aspect in terms of enhancement in the exchange performance due to restrictive resourceful behaviors. It is explained [47] that contracts are allied to the project performance in a way that the performance of any organization is dependent on the completeness of the contracts. Furthermore, contractual governance improves the project performance through effectively dealing with the project risks and uncertain situations and arranging the enforceable set standards with the project objectives [48]. Furthermore, it can be said that project governance also provides various methods by which the project goals can be set. Project governance provides resources, both to attain determined objectives and to monitor a determined performance as well. Consequently, prior literature such as that of [27] proved that project governance with its elements has a significant vital role in project performance. Researchers [17] strongly focused on governance elements to avoid uncertainties and risks (one of the important features of an effective governance mechanism) efficiently. Thus, the study proposed the following hypotheses:

**H1a:** *Fundamental elements have a positive impact on project performance.*

**H1b:** *Change elements have a positive impact on project performance.*

**H1c:** *Governance elements have a positive impact on project performance.*

**H2a:** *Fundamental elements have a positive impact on project management risk.*

**H2b:** *Change elements have a positive impact on project management risk.*

**H2c:** *Governance elements have a positive impact on project management risk.*

**H3:** *Project management risk has a positive effect on project performance.*

*3.2. Project Management Risk as Mediator*

Some research in IT has been conducted on project performance where risk management also played its intervening role [49]. However, the findings of some prior studies are controversial because these studies revealed that risk management has a not very high impact on project performance in terms of success [50]. Therefore, in the project atmosphere, we need some control tactics to have a strong impact on project performance concerning low intensities of the project risk, whereas we need reliance to make a stronger impact with respect to the high-risk situations [17]. Moreover, the consensus of previous studies on risk is that a double methodology is required which not only focuses on the negative perspective or a kind of threat but also on the fact that a positive perspective could be an opportunity [40]. Likewise, several studies have concentrated on the identification, evaluation, and minimizing of the factors of risk in IT-related projects, that studied the risk and concluded that risk can negatively impact the project performance of IT projects [3,51]. Additionally, former researchers also found that project-focused organizations used to avoid uncertainties and mitigated the risks by using effective governance structures. Therefore, these researchers have confidence in managing risk efficiently, which is an important aspect of an effective governance structure [17]. Thus, the study proposed the following hypotheses:

**H4a:** *Project management risk mediates the relationship between fundamental elements and project performance.*

**H4b:** *Project management risk mediates the relationship between change elements and project performance.*

**H4c:** *Project management risk mediates the relationship between governance elements and project performance.*

## 4. Research Method

### 4.1. Data Collection and Sample Size

The study samples include department heads, testers, analysts, designers/programmers, software engineers, and managers/team leaders from Pakistan-based software firms, collected through a convenience sampling technique. According to [12], by the middle of 2017 almost 320 software companies from all over Pakistan were listed as members of the Pakistan Software Houses Association for ITES P@SHA and IT. By considering the sample size, the study used a well-reputed and globally implemented sample size formula focusing on a finite population introduced by Krejcie and Morgan in 1970. By implementing this sampling formula, 175 out of 318 software firms were nominated randomly to be surveyed. Furthermore, 400 questionnaires were distributed to respondents from December 2019 to February 2020 through personal visits, email and an online survey to improve the generalizability of the findings. In response to the survey, 360 responses were returned, and after screening, 295 valid responses were usable, with a response rate of 73.75%.

### 4.2. Questionnaire and Measurements

A comprehensive literature review was undertaken to establish the observed items to assess the relationship between latent variables. The questionnaire was developed by adopting items from different studies and comprised 33 questions in six sections. Respondents had to evaluate their management on a Likert scale from 1 (strongly disagree) to 5 (strongly agree) on all item scales. In addition, change elements, fundamental elements and governance elements of contractual governance were measured with the items adopted from the study of [33,47] on a 5-point Likert scale of 1 (strong disagreement with the item) to 5 (strong agreement with the item). Furthermore, the 10 items were adopted from the work [14], to assess the project management risk. This instrument contains 07 items to determine the planning and control risk and 03 items to assess team risk ranging from 1 = strongly disagree to 5 = strongly agree. Moreover, to measure the construction of the project performance of the software development firm, the seven items were adopted from the work of [52]. The construct project performance (PP) was measured on two dimensions: process performance and product performance. A total of 05 items was constructed to evaluate the product performance, and 02 items were designed to measure the process performance.

## 5. Data Analysis and Results

We used Smart PLS 3.3.2 [53] (to assess the measurement and structural models following a two-step approach. The motivation of this research was to predict project performance as PLS is the most suitable analytical tool [54,55]. We also had a second-order measurement model with three dimensions, which suggested that Smart PLS would be the tool for the analysis. Since data was collected using a single source, to test the common method bias using the full collinearity method. The test showed that all the VIFs were lower than 5; thus, we can conclude that common method bias is not a major problem in our study.

### 5.1. Measurement Model

We followed the recommendations of [56], that confirmatory composite analysis (CCA) for assessing reflective constructs includes a process that involves an assessment of the item loadings, composite reliability, AVE, discriminant validity, homological validity, and predictive validity [56]. The loadings were all above 0.7, the AVE were higher than 0.5 and the CR was higher than 0.7 (see Table 1). Table 2 displays the discriminant validity assessment whereby in assessing the discriminant validity, the study followed the guidelines from [57], according to which the square root of the average variance extracted should be higher than the row and column values of the correlations. Our findings confirm that all the diagonal values exceeded the row and column values. Thus, we can conclude that the measures used in this are reliable, valid, and distinct.

**Table 1.** Description, Reliability, and Validity of the Measures.

| Constructs | Mean | Std. Dev. | Skewness | Kurtosis | Alpha | CR | AVE | Skewness | Kurtosis |
|---|---|---|---|---|---|---|---|---|---|
| Change Elements | 3.978 | 0.923 | −0.594 | 0.141897 | 0.836 | 0.875 | 0.703 | −0.48255 | 0.282859 |
| Fundamental Elements | 3.946 | 0.884 | 0.714 | 0.141897 | 0.829 | 0.884 | 0.718 | −0.26788 | 0.282859 |
| Governance Elements | 3.928 | 0.926 | −0.544 | 0.141897 | 0.794 | 0.874 | 0.698 | −0.39667 | 0.282859 |
| Project Management Risks | 3.925 | 0.871 | −0.698 | 0.141897 | 0.907 | 0.922 | 0.543 | 0.094179 | 0.282859 |
| Project Performance | 4.016 | 0.798 | −0.657 | 0.141897 | 0.833 | 0.859 | 0.552 | −0.20513 | 0.282859 |

**Table 2.** Discriminant Validity.

| Constructs | Change Elements | Fundamental Elements | Governance Elements | Project Management Risk | Project Performance |
|---|---|---|---|---|---|
| Change Elements | 0.838 | | | | |
| Fundamental Elements | −0.145 | 0.847 | | | |
| Governance Elements | 0.615 | −0.169 | 0.836 | | |
| Project Management Risk | 0.676 | −0.280 | 0.660 | 0.737 | |
| Project Performance | 0.734 | −0.251 | 0.618 | 0.691 | 0.743 |

*5.2. Structural Model*

The assessment of the structural model is based on multi-collinearity, path coefficient, significance, coefficient of determination ($R^2$), effect sizes ($f^2$) and predictive relevance ($Q^2$) [56]. All the VIFs were below 5; thus, multi-collinearity was not an issue. Next, the path coefficients, t-values and effect sizes are presented in Table 3. The value for $R^2$ should be equal to or greater than 0.1. We have two endogenous constructs in our model (see Figure 1) and the $R^2$ for the project management risk was 0.576 ($Q^2 = 0.297$), and the project performance was 0.704 ($Q^2 = 0.357$) which indicates that 57.6%, and 70.4% of the variance in the respective constructs can be explained by their predictors. $Q^2$ values greater than 0 indicate sufficient predictive relevance. The results in Table 3 show that the $R^2$ value is over 0.1, confirming the structural model's predictive capability. Furthermore, $Q^2$ establishes the predictive relevance of the endogenous constructs.

**Table 3.** Hypotheses Testing.

| Hypothesis | Relationships | Std Beta | Std Error | t-Value | *p*-Value | BCI LL | BCI UL | $f^2$ | Supported |
|---|---|---|---|---|---|---|---|---|---|
| H1a | FE→PP | −0.051 | 0.038 | 1.330 | *0.183* | −0.123 | 0.028 | 0.008 | No |
| H1b | CE→PP | 0.347 | 0.071 | 4.876 | *p < 0.001* | 0.207 | 0.482 | 0.204 | Yes |
| H1c | GE→PP | 0.065 | 0.059 | 1.101 | *0.271* | −0.047 | 0.181 | 0.007 | No |
| H2a | FE→PMR | −0.155 | 0.042 | 3.716 | *p < 0.001* | −0.243 | −0.079 | 0.055 | Yes |
| H2b | CE→PMR | 0.424 | 0.059 | 7.245 | *p < 0.001* | 0.313 | 0.545 | 0.262 | Yes |
| H2c | GE→PMR | 0.371 | 0.062 | 6.005 | *p < 0.001* | 0.243 | 0.485 | 0.202 | Yes |
| H3 | PMR→PP | 0.499 | 0.061 | 8.186 | *p < 0.001* | 0.379 | 0.616 | 0.353 | Yes |
| H4a | FE→PMR→PP | −0.078 | 0.024 | 3.251 | *p < 0.001* | −0.129 | −0.037 | | Yes |
| H4b | CE→PMR→PP | 0.212 | 0.042 | 5.041 | *p < 0.001* | 0.137 | 0.3 | | Yes |
| H4c | GE→PMR→PP | 0.185 | 0.035 | 5.33 | *p < 0.001* | 0.118 | 0.254 | | Yes |

| Endogenous Constructs | $R^2$ | $Q^2$ |
|---|---|---|
| PMR | 0.576 | 0.297 |
| PP | 0.704 | 0.357 |

Note: FE = Fundamental Elements; CE = Change Elements; GE = Governance Elements; PMR = Project Management Risks; PP = Project Performance.

Next, to assess the 10 hypotheses developed we ran a bootstrapping of 5000 resamples. Results are presented in Table 3 and Figure 2. First, we assessed the direct relationships before looking at the mediation effects. For H1, change elements were positively related to project performance (β = 0.347, t = 4.876, *p < 0.001*) but fundamental elements (β = −0.051, t = 1.33 *p = 0.183*) and governance elements (β = 0.065, t = 1.101, *p = 0.271*) were not significantly related to project performance. Hence, H1b was supported, but H1a and

H1c were not supported (see Table 3). For H2, the results revealed a significant relationship between fundamental elements and project management risk (β = −0.155, t = 3.716, *p < 0.001*), change elements and project management risk (β = 0.424, t = 7.245, *p < 0.001*), and governance elements and project management risk (β = 0.371, t = 6.005, *p < 0.001*), which supports H2a, H2b, and H2c. In addition, for H3, we found a significant relationship between project management risk and project performance β = 0.499, t = 8.186, *p < 0.001*). Hence, H3 was supported.

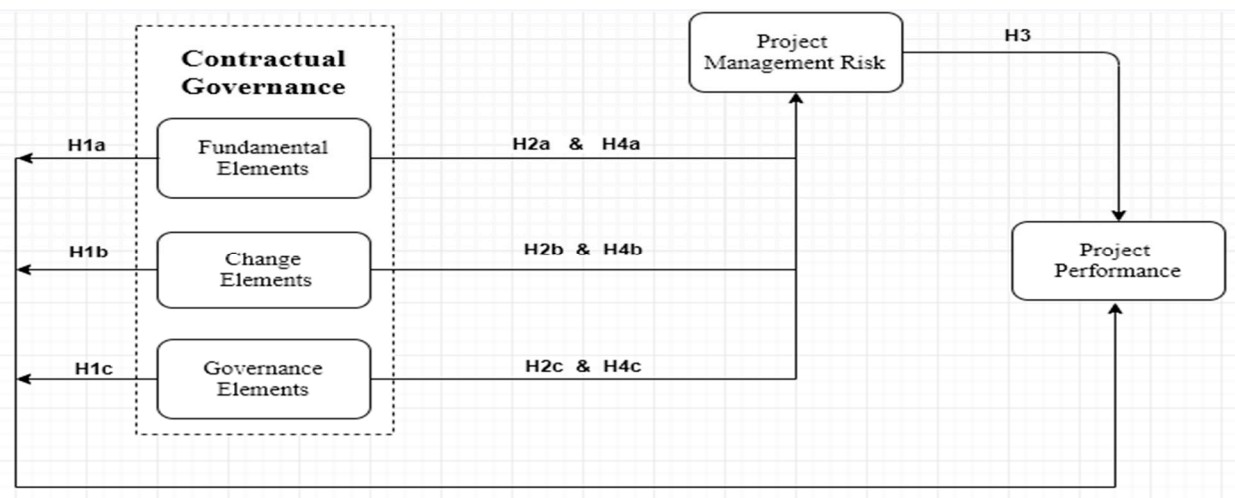

**Figure 1.** Research Model.

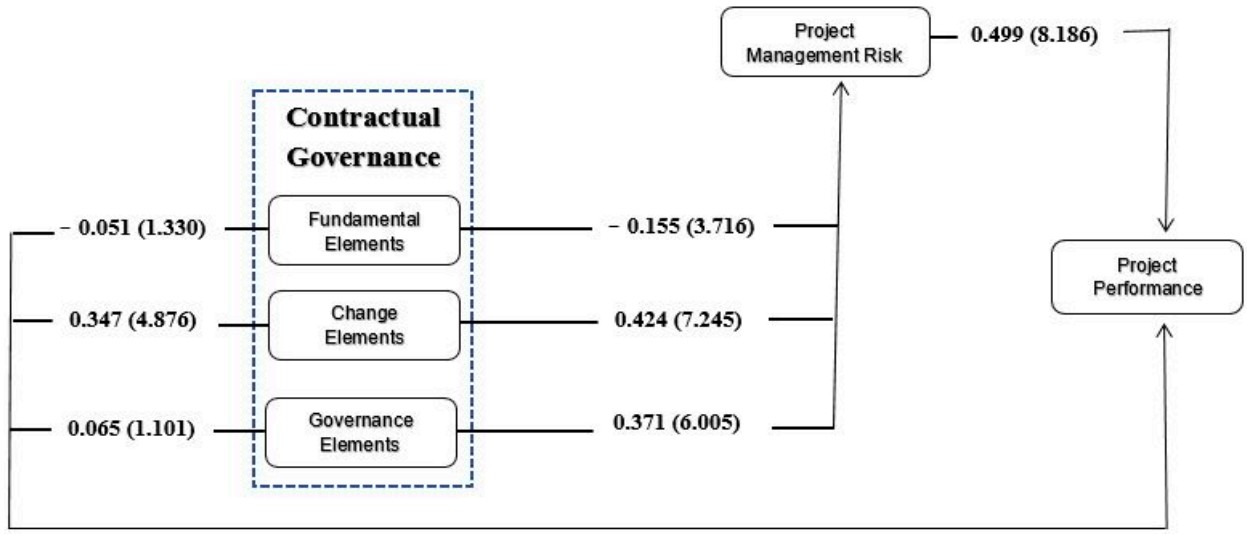

**Figure 2.** Path Model. Note(s): T values presented in parentheses.

To test the mediation effect, we used the bootstrapping indirect effect method (Preacher and Hayes, 2008) with a 5000 resample. The indirect effect of FE PMR PP (β = −0.078, *p < 0.01*, BCI LL = −0.129& BCI UL = −0.037), CE PMR PP (β = 0.212, *p < 0.01*, BCI LL = 0.137& BCI UL = 0.3) and GE PMR PP (β = 0.185, *p < 0.01*, BCI LL = 0.118 & BCI UL = 0.254) indicate that the indirect effect is statistically significant at the 0.01 level. This gives support to H4a, H4b and H4c of this study.

## 6. Conclusions and Implications

Project governance is one of the key features of project success and failure, parallel to other organizations. IT industries are also taking the initiative to realize how to select, develop and sustain governance and its supporting elements, through maintaining the

quality, and managing the risk in order to achieve the best performance of any project. In previous studies, the influence of contractual governance on project performance has been examined by a single item. Nevertheless, they focused primarily on studying the complementary aspects of governance mechanism. Thus, this study draws attention to developing a body of knowledge on contractual governance's three distinctive elements; that not only extends and expands the literature but also explains a key development of a mediation model to explore the effectiveness of these governance mechanisms in project management risk. Hence, it provides insights into project performance for IT sector practitioners.

### 6.1. Theoretical Contribution

The study contributed theoretically in multiple ways. Firstly, it provided an empirical description of principal−agent theory, establishing and testing a new mechanism between the elements of contractual governance (fundamental elements, change elements, and governance elements), project management risk, and project performance. Based on survey data collected from the software industry, the study's findings revealed a significant positive relationship between contractual governance and project performance, especially the fact that the change element plays an important role in boosting project performance. For example, H1b anticipated a substantial positive relationship of the CE with the PP; likewise, SEM results demonstrated that the CE is a positive significant predictor of project performance, respectively, which also endorsed the previous findings that claimed project governance is an important antecedent for the project's ongoing performance and also to fix the complications that arise in any project environment, by taking the necessary measures [11,27,46]. Moreover, the direct relationship of the contractual governance elements and the PMR that are specified in H2a, H2b, and H2c are significant and connoted with the findings of [12,16]. On the other hand, the hypothesis H3 is also supported, followed by the connection of the PMR to the PP and also confirmed the past studies [11,14].

Overall mediation results showed that PMR entirely mediated the connection between the FE, CE, GE, and PP. In addition, the FE, CE, GE and PP with the mediating role of project management risk indicated that H4a, H4b, and H4c are found to be significant; this corresponds with the studies of [12,16]. Our study revealed a better understanding and proper management of risk and its mitigation, and aimed to increase the performance of an IT project of agile mechanism through proper governance. Consequently, these results assisted in responding to the research questions and supporting the achievement of the objectives of this study. There are limited studies that have addressed the role of mediators in the relationship between project governance and project performance [17]; we have tested project risk management (PRM) as mediators [12]. Past research has rarely addressed the contractual governance as a multi-item construct, which is a great addition to extending the theory. Moreover, the findings also support the principal−agent theory, which accentuated the requirement of a clear-cut bond and contract between principal and agent. Furthermore, this study promotes the accountability of the principal−agent relationship which is the critical element of effective governance. Consequently, as project management entails a firm monitoring by the possessors, so a perpetual monitoring outcome is an enhanced project performance.

### 6.2. Practical Implications

Besides the theoretical implications, this study has many practical contributions relevant to the project governance mechanism to achieve the paramount performance of the project. It provides a substantial understanding of Pakistan's IT industry using agile methodology by addressing well-structured project contractual governance mechanisms for guiding and governing the projects towards successful outcomes. Additionally, this study has been conducted in an emerging market context where the IT industry has been emerging recently. The findings of this study may be applicable to other developing market's contexts. This study suggests the practical implications for IT project managers of

contractual governance elements such as fundamental elements, change elements, and governance elements, that could be applicable in other developing countries' projects to trigger their performance as well. Moreover, this research provides empirical evidence for the developing IT industry that effective project governance elements significantly positively affect project management. Likewise, they also work as the expedient tools for prompting the performance of IT projects. Therefore, this mechanism should also be worthwhile for the IT sector of any developing country to regulate their governance structure by solving the ongoing governance concentration issues and to attain pre-eminent performance. Specifically, in risk management process, ultimately to improve project performance there should be proper technical procedures to focus on and to sustain it throughout the project, to satisfy the consumers of IT projects and to uphold the market value. Furthermore, the potential risks should also be identified systematically by the practitioners, managed and tackled in a timely manner to minimize their eventual threats. Moreover, the concepts of principal−agent theory will be helpful for the practitioners to develop a governance structure, based on accountability that should be applicable in every step of the project's life cycle, and they must be properly followed to generate long-term progressive contracts. Additionally, it explains the abilities of the concerned sector by taking various strategies for IT practitioners to achieve and then enhance their project performance.

*6.3. Limitations and Future Research*

A limitation of this study stems from the use of quantitative approach that did not explain the worth of this in a studied relationship; therefore, further research should investigate this model with a qualitative approach to explore new ideas. Second, the study only investigated the impacts of contractual governance on project performance through intervening concepts of project management risk by collecting data from the IT sector; future research should be conducted by collecting data from the other sectors of the economy, as well as in other developing countries with the same research model at various organizational levels. Third, future studies should also use the elements of contractual governance such as fundamental elements, change elements, and governance elements as separate variables to the current model. Fourth, this study should be extended by involving "project quality" as a moderator or mediator in the current research model. Furthermore, distributed decision-making theory should also be used in combination with principal−agent theory for future research.

**Author Contributions:** Conceptualization, A.S., M.M. and M.U.S.; methodology, A.S., M.M. and M.U.S.; software, A.S. and M.U.S.; validation, A.S., M.M. and M.U.S.; formal analysis, A.S., M.M. and M.U.S.; investigation, A.S., M.M. and M.U.S.; resources, A.S., M.M. and M.U.S.; data curation, A.S. and M.U.S.; writing—original draft preparation, A.S., M.M. and M.U.S.; writing—review and editing, A.S., M.M. and M.U.S.; supervision, M.M.; project administration, A.S.; funding acquisition, M.M. All authors have read and agreed to the published version of the manuscript.

**Funding:** This research received no external funding.

**Data Availability Statement:** It is available on request from the first Author.

**Acknowledgments:** All authors are very obliged to those anonymous judges for their support, suggestions and appreciated remarks.

**Conflicts of Interest:** The authors declare no conflict of interest.

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
