# Peer review of "Effects of Contractual Governance on IT Project Performance under the Mediating Role of Project Management Risk: An Emerging Market Context"

_information, doi:10.3390/info14090490_

Round 1

Reviewer 1 Report

The topic of the paper is good. The following comments to the authors, please revise carefully.

1. The literature review should be enhanced, especially on project management risk, fundamental elements. These articles may be helpful for improving this paper: two-level principal-agent model for schedule risk control of IT outsourcing project based on genetic algorithm, a bilevel whale optimization algorithm for risk management scheduling of information technology projects considering outsourcing, simulated annealing genetic algorithm based schedule risk management of IT outsourcing project.

2. The quality of Table 1-3 need to be improved. The notes below Table 3, the punctuation mark between CE and GE should be a semicolon instead of a comma.

3. Some hypotheses are put forward in 2.5, which is suggested to be shown in another section.

4. Read the sentences in section 3.2, and correct the mistakes of spelling.

5. Draw some figures will be helpful for understanding the analysis in section 4.

Minor editing of English language required

Author Response

Dear Reviewer,

Thank you for your time and support to improve our paper. We have revised the paper following your recommendations as well as the recommendations of the other reviewer. Please find in the attached file the response.

Thank you again for your support.

regards,

Muhammad Mohiuddin

Laval University, Quebec, Canada.

Reviewer 2 Report

The authors present a very interesting paper regarding the Effects of contractual governance on IT project performance: Mediating role of project management risk.

Some improvements are suggested:

1-   In the abstract (and across the paper) is not clear why the authors conducted this research. What is the background of this study? What is the problem that this research is aiming to tackle?

2-   The authors write about IT project performance. However there is no mention to how the majority of IT projects are run today – Agile (scrum, etc..). Shouldn’t this topic be part of the research? Can the authors connect the research conducted with the agile methodology in the IT industry?

3-   The references must be adjusted to the journal standards.

4-   The authors should also review the number of references. It should not exceed the number between 40-60.

5-   Furthermore the majority of references are older than 10 years. This is not a good approach namely when presenting a research in the IT field. The authors should take this point in consideration.

Thank you

good job

Author Response

(The authors gave the same response as above.)

Round 2

Reviewer 1 Report

1.      Please mark all change in the manuscript in a different color. Now it is difficult to find them and unserstand.

2.      The literature (Lu et al., 2015), which was cited many times. However it could not be found in the reference list.

Minor editing

Author Response

Dear Reviewer,

Thank you for your valubale comments. We have revised the Abstract and went through the whole paper and corrected where necessary. Please find the version with track change of the paper attached here.

Regarding the REFERENCE you mentioned, in fact, that reference is there in the reference section.

Thank you again for your supports.

With warmest regards,

Muhammad Mohiuddin 

Reviewer 2 Report

Well done!

Round 3

Reviewer 1 Report

I have nomore questions about the paper.

minor